# Impact of local mask mandates upon COVID-19 case rates in Oklahoma

**Jared D. Taylor** [1,2]*, **Melinda H. McCann**[3], **Scott J. Richter**[4], **Dakota Matson**[5], **Jordan Robert** [5]

**1** Oklahoma State Department of Health, Oklahoma State University, Oklahoma City, Oklahoma, United States of America, **2** Oklahoma State University Department of Veterinary Pathobiology, Stillwater, OK, United States of America, **3** Oklahoma State University Department of Statistics, Stillwater, OK, United States of America, **4** Statistical Consulting Center and Mathematics and Statistics Department, University of North Carolina, Greensboro, Greensboro, NC, United States of America, **5** Oklahoma State Department of Health, Oklahoma City, Oklahoma, United States of America

* jared.d.taylor@okstate.edu

**Data Availability Statement:** Summary data are held in Figshare (https://figshare.com/articles/dataset/Mask_Mandate_rates_by_day_xlsx/19834840). Initial data utilized to create case

## Abstract

Use of face coverings has been shown to reduce transmission of SARS-CoV-2. Despite encouragements from the CDC and other public health entities, resistance to usage of masks remains, forcing government entities to create mandates to compel use. The state of Oklahoma did not create a state-wide mask mandate, but numerous municipalities within the state did. This study compares case rates in communities with mandates to those without mandates, at the same time and in the same state (thus keeping other mitigation approaches similar). Diagnosed cases of COVID-19 were extracted from the Oklahoma State Department of Health reportable disease database. Daily case rates were established based upon listed city of residence. The daily case rate difference between each locality with a mask mandate were compared to rates for the portions of the state without a mandate. All differences were then set to a d0 point of reference (date of mandate implementation). Piecewise linear regression analysis of the difference in SARS-CoV-2 infection rates between mandated and non-mandated populations before and after adoption of mask mandates was then done. Prior to adopting mask mandates, those municipalities that eventually adopted mandates had higher transmission rates than the rest of the state, with the mean case rate difference per 100,000 people increasing by 0.32 cases per day (slope of difference = 0.32; 95% CI 0.13 to 0.51). For the post-mandate time period, the differences are decreasing (slope of -0.24; 95% CI -0.32 to -0.15). The pre- and post- mandate slopes differed significantly (p<0.001). The change in slope direction (-0.59; 95% CI -0.80 to -0.37) shows a move toward reconvergence in new case diagnoses between the two populations. Compared to rates in communities without mask mandates, transmission rates of SARS-CoV-2 slowed notably in those communities that adopted a mask mandate. This study suggests that government mandates may play a role in reducing transmission of SARS-CoV-2, and other infectious respiratory conditions.

counts by municipality will not be available to avoid disclosure of confidential medical information. Data restrictions were imposed by OSDH IRB. For further information, contact all of the following: Evaren Page (EvarenP@health.ok.gov); Brandon Wesbury (brandon.wesbury@health.ok.gov); and Kevin McMahon (Kevin McMahon (kevin.mcmahon@health.ok.gov).

**Funding:** The authors received no specific funding for this work.

**Competing interests:** No authors have competing interests.

# Introduction

Efforts for reducing community transmission of severe acute respiratory syndrome Coronavirus 2 (SARS-CoV-2), the causative agent for Coronavirus disease 2019 (COVID-19), initially focused on dramatic measures including federally-endorsed and state-mandated "shutdowns." These included closure of schools (primary, secondary, and higher education), businesses, churches, and events, as well as restrictions on travel and nearly all commercial and non-commercial activities outside the home. These restrictions were effective in slowing the emergence of the pandemic in the US [1], but inflicted great costs, economic and otherwise [2, 3]. With the recognition that these measures could not be permanent, and as understanding of transmission increased, more targeted policy recommendations became common, including reduction of large gatherings, encouragement of social distancing, utilization of face coverings [4], and eventually vaccination. Once the shutdowns were eased transmission increased, particularly in areas of the US where cases had been low previously. This may be attributed to limited adoption of the targeted mitigation steps, which was likely driven by myriad of factors, including politicization of such measures as well as the economic burden of social distancing efforts. However, it also calls into question the effectiveness of the suggested mitigation strategies.

Of particular interest has been the effectiveness of use of masks or face coverings by the general public. Recommendations regarding use of masks or face coverings have varied since the beginning of the epidemic. Initially, wearing of face coverings was advised only for symptomatic individuals or in healthcare settings. Gradually, evidence suggested transmission occurred primarily via respiratory secretions rather than via fomites, leading to recommendations for face coverings to be more commonly employed [5]. Numerous studies have demonstrated the efficacy of face coverings in reducing transmission [6, 7], including a review which thoroughly examined the evidence on mask usage [8]. The authors recommended adoption of public cloth mask wearing, in conjunction with other measures. However, they also acknowledge challenges in evaluating efficacy of seeking population-wide mask usage, including deficiencies in compliance. Efficacy of masks were also called into question by Shah, et al. [9], who found that more commonly used cloth masks and even surgical masks offer relatively low apparent filtration efficiencies (<15%). Cheng, et al. [10] examined case incidence in populations with community-wide masking as compared to that of non-mask wearing communities, but this was across different countries where other societal, meteorological, or regulatory factors may have also played a role in reducing transmission. Importantly, this study compared the Hong Kong Special Administrative Region (HKSAR) to European and North American countries. The authors acknowledge that the HKSAR general population was on high alert after the previous SARS epidemic, and documented compliance of face mask usage of >95% on three consecutive days.

Compliance comparable to that reported by Cheng, et al. would not be expected in the US in the absence of government-imposed mask mandates, and perhaps not even then. Thus, one must be careful in considering what benefit could be derived from mask mandates in the US. To this question, incidence was assessed within a single US state before and after mitigation measures [11]. This examination included multiple mitigation steps, not just face covering mandates. Moreover, it was a simple longitudinal assessment, which can be confounded by changes over time due to other causes. A more recent project by Lyu and Wehby examined case occurrence data in light of statewide mask mandates or employment-related mask mandates, and in doing so attempted to account for shutdown restrictions in the states examined as well as week of the year [12]. A notable limitation of this work is the fact they only examined results over a seven week timeframe. Periodic "waves" of cases in various states or geographic

regions has been a recurring phenomenon, but these waves have not necessarily followed seasonal or recurring intervals and are often difficult to predict or explain [13]. The impact of social norms, messaging, and now vaccination efforts vary enormously across states, and may account for some of this periodicity. Regardless of cause(s), it is unlikely that the complexities of this periodicity could be captured in any seven week period. A publication by Joo, et al., compared case data at the county level for states with mandates to those without mandates and described a reduction in hospitalization rates following state-wide mask mandates [14], over a roughly seven month period. Both Joo, et al. and Lyu and Wehby studies are limited by the fact that it is difficult or impossible to account for across-state variables, as well as changes in transmission over time due to other external factors. Specifically, the models included state as a variable in the regression. However, when no replicate exists for each state, such inclusion requires that the effect of the state is constant across the entire study period. This assumption cannot be evaluated and may be flawed. These factors complicate efforts to compare mitigation steps across states, and particularly over relatively short periods of time (as Lyu and Wehby examined).

Lyu and Wehby's model included population density, socioeconomic status, and demographic information. Such information may strengthen the model in many ways, but it assumes that the effect of these considerations are consistent across multiple states. It is further likely that the prevailing political and social norms within a state are at least as important in influencing opinions and actions, as are the effects of socioeconomic or racial characteristics [15]. As such, nations, states, or even smaller municipalities that are considering mandating one or more mitigation strategies may prefer an examination of impacts of individual mitigation steps (as opposed to multiple policies concurrently), as well as comparisons within a state, to determine efficacy of mask mandates. Lyu and Wehby used an event study approach, and examined discreet time intervals following adoptions of mandates. This, or similar approaches (such as the simpler variation on a "difference in differences" approach reported here), may be best suited to account for variability over time.

Similar to most states in the US, Oklahoma instituted various state-wide restrictions early in the pandemic. These were initiated on March 24, with a comprehensive shut-down order issued on April 1. On April 22 the governor announced his "Open Up and Recovery Safely (OURS)" plan which involved various dates and stages for the state to reopen. By June 1, all state-wide restrictions were lifted and none were reimposed before late November 2020. Despite this absence of statewide restrictions, numerous municipalities instituted mask mandates. These municipalities are generally larger population centers in the state, including the two largest metropolitan areas- Oklahoma City and Tulsa- along with many of their surrounding suburbs. However, there are a number of municipalities with populations of 15,000 to 50,000 that are not a directly related suburb of Oklahoma City or Tulsa, as well as handful of non-suburban municipalities with populations of less than 10,000. These mandates were instituted in two general waves, with some variation in each of those. The first wave ranged from late March to early September, with most taking effect in July (Table 1). The second wave began in early November and continued through mid-December. The study described here analyzed the effect of mask mandates on case density rates. Rather than simply comparing within a location longitudinally, the primary goal was to compare populations with a mask mandate to those without a mandate at the same time point. Because mandates went into effect at different times, it was decided to use a modified difference-in-differences approach for comparing municipalities with mandates to those without. This entailed establishing a universal day 0 (d0) for mask mandate implementation and comparing all citizens residing in a given municipality with a mask mandate on that day to all citizens in the state in a municipality without a mask mandate. The hypothesis was that mask mandates would have no impact on case density rates.

**Table 1. Oklahoma municipalities that adopted mask mandates, along with the population, mandate start and expiration dates.**

| City | Population | Mandate Start | Mandate End |
|---|---|---|---|
| Altus | 18,338 | 3/23/2020 | 5/4/2021 |
| Guthrie | 11,376 | 4/7/2020 | 5/5/2020 |
| Chickasha | 16,337 | 4/10/2020 | 5/1/2020 |
| Anadarko | 6,504 | 4/18/2020 | 4/14/2021 |
| Ada | 17,235 | 4/20/2020 | 5/17/2021 |
| Norman | 124,880 | 7/7/2020 | 5/18/2021 |
| Stillwater | 50,299 | 7/11/2020 | 5/25/2021 |
| Tulsa | 401,190 | 7/12/2020 | 4/30/2021 |
| Oklahoma City | 655,057 | 7/17/2020 | 3/5/2021 |
| Lawton/Ft. Sill | 93,025 | 7/20/2020 | 3/23/2021 |
| Warr Acres | 10,118 | 7/21/2020 | 3/31/2021 |
| The Village | 9,564 | 7/22/2020 | 5/1/2021 |
| Spencer | 3,968 | 7/23/2020 | 12/31/2021 |
| Edmond | 9,4054 | 7/27/2020 | 3/23/2021 |
| Nichols Hills | 3,938 | 7/27/2020 | 4/30/2021 |
| Shawnee | 31,436 | 7/27/2020 | 4/30/2021 |
| Midwest City | 5,7407 | 7/28/2020 | 3/31/2021 |
| Del City | 21,822 | 8/3/2020 | 8/31/2020 |
| Tahlequah | 16,819 | 8/3/2020 | 3/31/2021 |
| Choctaw | 12,474 | 8/5/2020 | 4/20/2021 |
| McAlester | 17,814 | 8/20/2020 | 11/30/2020 |
| Okmulgee | 11,846 | 11/9/2020 | 4/14/2021 |
| Jenks | 23,767 | 11/11/2020 | 4/30/2021 |
| Ardmore | 24,698 | 11/12/2020 | 4/5/2021 |
| Clinton | 9,217 | 11/17/2020 | 3/19/2021 |
| Sapulpa | 21,278 | 11/18/2020 | 5/4/2021 |
| Grove | 6,957 | 11/20/2020 | 4/6/2021 |
| Hominy | 3,431 | 11/20/2020 | 4/14/2020 |
| Okemah | 3,178 | 11/23/2020 | 4/12/2021 |
| Chouteau | 2,066 | 11/24/2020 | 6/7/2021 |
| Muskogee | 37,113 | 11/25/2020 | 1/25/2021 |
| Sand Springs | 19,905 | 11/27/2020 | 4/27/2021 |
| Enid | 49,688 | 12/3/2020 | 3/17/2021 |
| Ponca City | 24,134 | 12/14/2020 | 4/12/2021 |
| Seminole | 7,219 | 12/14/2020 | 4/14/2021 |
| Vinita | 5,423 | 12/15/2020 | 2/17/2021 |
| Glenpool | 13,936 | 12/18/2020 | 4/5/2021 |
| Claremore | 18,753 | 12/20/2020 | 4/5/2021 |
| Purcell | 6,401 | 12/21/2020 | 3/1/2021 |

## Materials and methods

Throughout the study period, SARS-CoV-2 testing was only available from healthcare settings (i.e., no tests were approved and marketed for at-home testing during the study period). Moreover, SARS-CoV-2 was a reportable disease in Oklahoma, with all healthcare providers and diagnostic laboratories required to report positive cases to the Oklahoma State Department of Health (OSDH), which there then recorded into the OSDH Public Health Investigation and Disease Detection of Oklahoma (PHIDDO) system. For this study, the OSDH PHIDDO

records were queried for all diagnosed cases of COVID from March 17th, 2020 to March 1st, 2021. This included all cases diagnosed via polymerase chain reaction (PCR) testing (classified per criteria proposed by the Centers for Disease Control and Prevention as Confirmed Cases), antigen detection testing (classified as Probable Cases) and cases where a person had a known exposure to a COVID infected individual and subsequently developed symptoms consistent with COVID (designated as epidemiological links, and classified as Probable Cases) [16]. Cases were examined for city or town of residence and event date. Event date is the date of symptom onset if the person has symptoms, or date of sample collection if the person either did not demonstrate symptoms, or was sampled prior to symptom onset. Data were collected as part of routine reportable disease surveillance activities, and were deidentified prior to analysis. The study design was reviewed by the OSDH Institutional Review Board (IRB) administrator, who determined that the procedures are considered public health practice and further IRB review/oversight was not required.

Populations for towns of interest were derived from 2019 census estimates [17]. Date of implementation of mask mandates were obtained from websites compiling municipalities with mandates [18], through communication with public health officials throughout the state, and confirmed via search of city websites or other public records sources. Subsequently, city or town officials were contacted to confirm the adoption and effective dates of the mask mandate.

Daily case density rates (cases per 100,000 people) were calculated for each municipality that established a mandate, including calculation of rates for time points prior to implementation (designated d1, d2, etc.). Additionally, corresponding rates were determined for the rest of the state for the same date, excluding other municipalities that already had mandates in effect on that date. The difference in rates was then calculated by subtracting the rate for the "non-mandated regions" of the state from the rate for the municipality in question. A parallel trends analysis was done to confirm suitability of using a modified difference-in-differences approach. This entailed plotting case rates for municipalities that would adopt a mandate and those municipalities that did not, by date (without regard to date of implementation of mandates).

After the difference in rates were calculated for the 39 municipalities with a mask mandate, all were aligned to a "day 0" (d0) set point, and curated to include a range from d-45 to d90 (or as close to that range as possible, as some municipalities had instituted their mandate less than 90 days prior to analysis or the mandate was in effect for less than 90 days). For municipalities that had a mandate that expired within the period of analysis, effective at the time of expiration/termination the population for that municipality was removed from the state's population estimate and the cases for that municipality were removed from all calculations. This prevents contamination of results of "mandated" communities by a municipality without a mandate in effect at the time, and also prevents counting cases and population as "non-mandated" in the period immediately following a previous mandate.

Comparisons were then done within a pre-mandate time period (ranging from d-45 to d0) and post-mandate (d14 to d90). No examination was done of d1 to d14, to account for the transition from pre- to post-mandate. Descriptive statistics were calculated for pre- and post-mandate periods. The difference in rates for each day was determined to be either less than or greater than zero (i.e., higher or lower in mandated municipalities vs. non-mandated municipalities [or municipalities that would eventually adopt a mandate, in the case of pre-mandate time period]). The collection, organization, and curation of data is summarized in Fig 1.

To assess statistical significance of the differences found, a piecewise linear regression model was fit to the difference in case rates and a test was performed to determine if the slopes of the pre- and post-mandate periods were equal. A Durbin-Watson test was also performed to determine if serial correlation was an issue, and showed significant serial correlation.

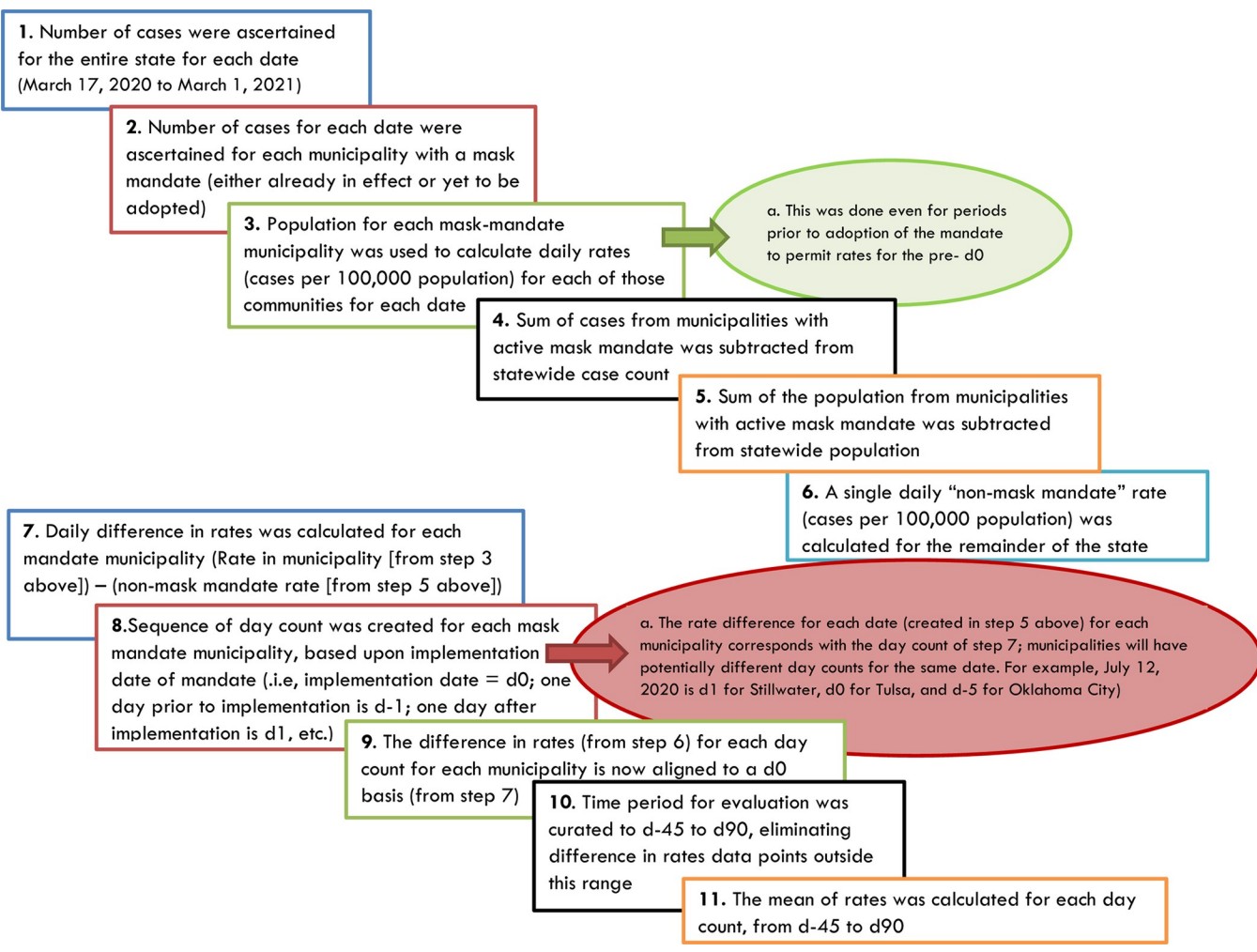

**Fig 1. Summary of data collection, organization, and curation process for comparing case rates between municipalities with mask mandates vs. those without mandates.**

Consequently, a one-lag moving average term was added to the model, and the Durbin-Watson test for this adjusted model showed no evidence of serial correlation (p = 0.864). Additionally, residual plots from this modified model were obtained to examine possible departures from the required assumptions and 95% confidence intervals were calculated for slopes for pre-mandate rate differences, as well as post-mandate rate differences. Analysis was repeated with exclusion of apparent outlier observations to see if their inclusion altered conclusions.

## Results

The search process identified 39 municipalities as issuing mandates. A listing of those municipalities is provided in Table 1.

The "non-mandated" population ranged from a maximum of approximately 3,957,000 (i.e., the full state population) to approximately 1,994,000. When assessed strictly by date, parallel trends were clearly demonstrated between mandated and non-mandated communities, meaning assessment of difference-in-differences was appropriate (S1 Fig). Residual plots of data after setting to d0 showed no evidence of assumption violations, and no outlier observations

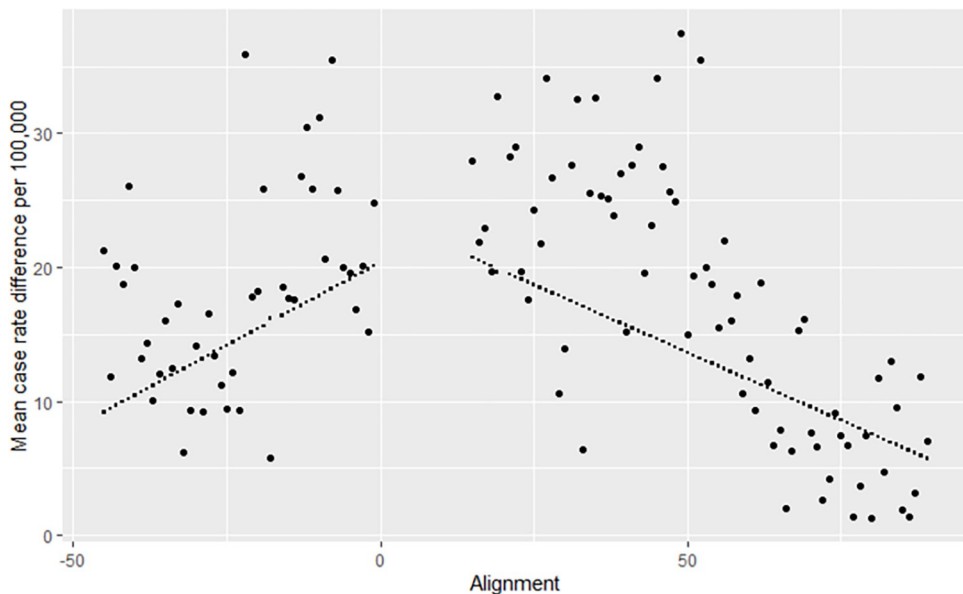

**Fig 2. Difference in case rates per 100,000 population, between municipalities with mask mandates vs. those without mandates.**

exerted disproportionate impact; therefore, all data points were retained in final results. Fig 2 shows a plot of the resulting model for the case rates. The pre-mandate time period showed rapidly increasing case rate differences between the locations that were to enact a mask mandate and the rest of the state (slope of 0.32; 95% CI 0.13 to 0.51). In other words, case rates in those areas were increasing faster than in other parts of the state, with the mean case rate in municipalities that will adopt a mandate increasing by 0.32 cases per 100,000 people per day (as compared to the rest of the state). For the post-mandate time period, the differences are decreasing (slope of -0.24; 95% CI -0.32 to -0.15, reflecting that changes in daily case rates were now 0.24 cases per 100,000 lower in communities that adopted mandates (as compared to the rest of the state). The pre- and post- mandate slopes differed significantly (p<0.001). The change in slope direction (-0.59; 95% CI -0.80 to -0.37) shows a move toward reconvergence in new case diagnoses between the two populations. Thus, while overall rates remained higher in mandate communities, growth in rates had slowed and were approaching comparability to non-mandated areas.

## Discussion

Early state-wide restrictions coincided with very low counts of COVID-19 cases. Only a few, relatively small municipalities instituted mask-mandates during the period of state-wide restrictions. Case counts throughout the state increased after the state relaxed restrictions and permitted greater return to normal activities. However, restrictions were lifted in the summer, at a time where transmission was somewhat dampened by meteorological conditions and/or other factors (Fig 3). Nonetheless, the major metropolitan areas saw case count increases, and adopted mandates in mid-July. This was followed by mandates in many surrounding suburban communities. Despite these measures, case counts across the state increased notably as fall conditions emerged and schools returned to classes. Case counts (as well as hospitalizations and deaths) increased greatly in late fall and early winter (Fig 3), resulting in the second wave of mandates in November and December. The method used in the analysis reported here (a

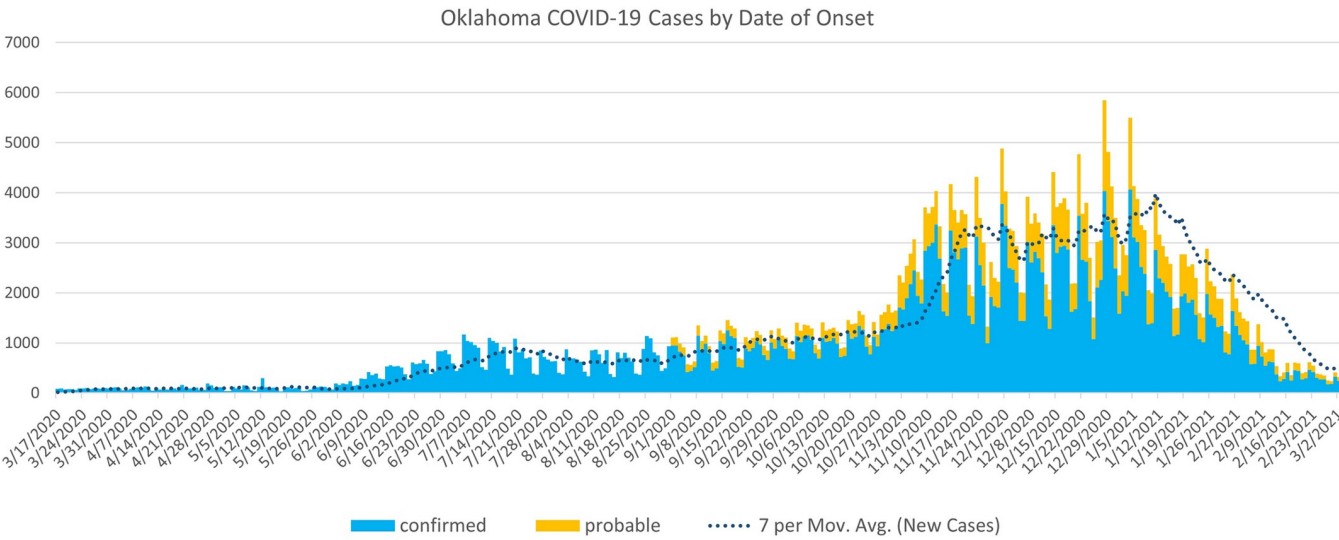

**Fig 3. Daily case numbers and 7 day average of cases for the state of Oklahoma.** Source: OSDH Weekly Epidemiology and Surveillance Report.

modification of the difference-in-differences approach, setting to a d0 implementation period) allows inclusion of this second wave of mask mandates in an analysis of efficacy of these measures, despite being implemented at notably different times than the first wave. While accurately reflecting statewide disease burden, these general descriptions of case rate dynamics obscure diversity of changes across the state. Our analysis shows that, prior to adoption of mask mandates, rates were increasing faster in the areas that would eventually adopt mandates than in those that would not (i.e., the difference in rates was greater than zero). After communities began adopting mask mandates, case rates became much less divergent in mandate and non-mandate communities, albeit they did not return to equal in the time period examined. Despite the difference in case rates remaining greater than zero, the change in slope of the difference is dramatic and clinically significant.

Results reported here show that implementation of mask mandates coincided with a decrease in transmission of COVID-19 within those communities implementing the mandate. Prior to mandates being adopted, rates of transmission were much higher in communities that would eventually adopt mandates, as compared to the remainder of the state (i.e., the difference between rates was greater than zero, and had a clear discernible positive slope). It could be argued that the higher rates of disease prior to mandates point to inherent differences between those communities and others that did not adopt mandates. The most apparent of these differences would seem to be degree of urbanization. While relatively little research is available on COVID-19 transmission in rural vs. urban areas, increased population density has been shown to favor disease propagation [19]. The municipalities adopting mask mandates were generally larger than communities not adopting mandates, and include not only the two largest major metropolitan areas (Oklahoma City and Tulsa), but eight of the ten largest population centers of the state. Much of the remainder of the state is more rural. The fact that mandates appeared to largely mitigate this difference (which was manifest prior to d0) is supportive of the efficacy of mask mandates. There are several possible explanations for the persistently higher case rate difference between communities with mask mandates and those without, even after mandate implementation. The most dramatic increase in case rates initiated the introduction of many of the mandates, and continued for some time following them. This period coincided roughly with colder ambient conditions, and may reflect the limitations

of mask mandates to overcome increased transmissibility in urban environments. Alternatively, it may reflect gradual non-compliance with the mandate, as people tired of mitigation efforts. Finally, it may reflect something of a spillover effect, where the benefits of mask mandates are not simply limited to those localities with the mandates but also impact transmission in other areas [20].

The study reported here employs a simpler analytical approach but may offer several advantages compared to previous work examining the effectiveness of mask mandates. The authors are unaware of any other studies that have compared communities with and without mandates within the same state. This approach avoids complications of varying social norms and attitudes, as well as other governmental restrictions or mitigation efforts. Moreover, the approach of calculating rate differences between municipalities with mandates versus the remainder of the state, and then setting all implementation dates to a standard d0 basis avoids any potential confounding effects of seasonal variation and changes over time that have been seen throughout the country and world with COVID-19. If utilization of masks is effective in preventing COVID-19 transmission, a compounding effect would be expected to occur over time, as secondary cases that would have developed are reduced by elimination of primary infections. However, without the d0 approach used here, this could not be appreciated unless all mandates were implemented on the same day. Our study also allows examination in one analysis of disease transmission dynamics between two general time periods of mandate adoption (mid- to late summer vs. late fall/early winter).

There are several limitations of the study reported here. The first is the observational nature of the study, which precludes determination of cause and effect. Most reported studies on mask usage impact on COVID-19 transmission have been observational, and our results are consistent with the majority of them. One randomized control trial has been reported, which found no benefit to wearing a mask in preventing COVID-19 infection [21]. Additional research is warranted to determine if this discrepancy is due to differences in prevalence at the time of study, compliance, or other factors; or, whether mask mandates or even voluntary use of masks corresponds with increased awareness and adoption of other mitigation strategies.

The current study is also ecological in nature. It was not possible to assess compliance with mask usage in communities with or without mandates, and there is no ability to determine whether cases were occurring disproportionately among individuals wearing masks. Enforcement of mandates varied notably across the various municipalities and even over time. Given that many of the mandates were limited to indoor gatherings, enforcement was done via business or venue staff, or at least was dependent upon staff notifying law enforcement of failure to comply [22]. Thus, it is also impossible to determine that mandates increased utilization of masks. Nonetheless, these limitations do not reduce the relevance of this study for policy makers who are most concerned with outcomes at the population level.

An additional limitation for the present study includes the need to rely on diagnosed cases. The d-45 period for early adopting cities aligned with the time of lifting of state-wide restrictions. However, this time also coincided with increasing access to widespread testing. As such, it is important to recognize that any noted increases may reflect increased detection as well as increased transmission. Specifically, if case rates increased in locales after mandate implementation, it should not be assumed that the mandates failed. We feel that this concern is limited by the approach used for our analysis, including comparison of communities at the same time (rather than comparing communities to themselves on a "before and after" basis). There would be no such concerns for municipalities in the second wave, as COVID-19 testing was not limited in availability during the fall and winter. Additionally, it is known that diagnosed cases rates do not account for all disease transmission. The relative efficacy of detection of a disease can be assessed through percent positivity, which refers to the percentage of all tests

performed that are positive. Robust and effective surveillance efforts can be assumed to be present only if a relatively large proportion of people needing to be tested have access to the test. This would include anyone with symptoms consistent with COVID-19, as well as those with a history of exposure, those at high-risk of exposure, etc. When such widespread availability of testing is present, many of the results should be negative. In this instance, some of the people being tested have some other explanation for their symptoms, while others who were potentially exposed are not infected (or at least not shedding virus yet). When percent positivity is relatively high, it is likely that testing is only being performed on people who are highly symptomatic or most likely to be infected. In this case, subclinical and pre-clinical infections are likely being missed. Oklahoma maintained a relatively high percent positivity rate throughout the study period (always >5% and typically >10%) [23]. This indicates that a relatively large proportion of infections were not being detected in the state. We know of no reason why there would be a bias in percentage of infections diagnosed between communities with mask mandates and those without, but it cannot be ruled out as a potential bias as accurate and comprehensive percent positivity is not available at a city or county level in Oklahoma.

The study reported here includes a timeframe (late December 2020 to March 1, 2021) that coincided with introduction of SARS-CoV-2 vaccines. During this period, vaccination efforts were focused on high-risk situations (residents and employees at long term care facilities and healthcare providers), with only a small percentage of the state population becoming fully vaccinated before the termination of the study. It is therefore unlikely that findings of our study are biased in any way due to vaccination. It is unclear what impact widespread vaccination or immunity would have on the findings; thus, further study is warranted. Finally, more complicated methods exist that attempt to account for other variables, including population density, and socioeconomic or demographic characteristics. We chose to employ a simpler approach for a number of reasons. First, while population density is assuredly associated with increased risk of disease, inclusion of population density as a covariate in our study would be difficult. Specifically, the largest metropolitan areas adopted mask mandates, and almost no rural areas adopted mask mandates. Only in the middle range of population density do we have the ability to compare within roughly equal sized communities. Extrapolating this to the high and low ranges of population density would risk violating unverifiable assumptions. Moreover, inclusion of other variables, including socioeconomic status and demographic information (at the municipal level) adds complexity to the model and is only valuable if it is predictive of risk of infection. Applying additional variables, including socioeconomic status and demographic information at the ecological level risks assumptions of association that cannot be tested or verified. Without adequate empirical data to confirm associations of these variables with the outcome of interest, we were reluctant to unnecessarily complicate the model.

The CDC previously issued guidance that fully immunized people do not need to wear face coverings or practice social distancing [24]. However, this was later revised to state that masking may be indicated, even for vaccinated individuals, in circumstances of high community spread. Vaccine uptake slowed throughout the US after initial offerings, and vaccination efforts in other parts of the world also lag far behind what will be needed to contain the virus. In addition, the emergence of variants, uncertainty of duration of immunity, and vaccine hesitancy all suggest achieving "herd immunity" could be difficult, and impacts in the interim may be devastating. It remains to be seen whether seasonal patterns will become evident, with return of high numbers of cases in winter; seasonality of COVID-19 remains undetermined, as evidence is contradictory [25–27]. In such cases, policy makers may again need to examine what measures (voluntary or compulsory) are needed to protect public health. Indeed, recent work has suggested that improved compliance with mitigation strategies could be even more critical than vaccine efficacy, at least in terms of short term outcomes of hospitalizations and

deaths [28]. As such, it is critical that states and municipalities be able to assess the impact of mask mandates on community transmission. The results reported here demonstrate that local mandates are indeed associated with reducing case densities. This evidence may also prove beneficial in considering mitigation strategies for future infectious disease outbreaks.

## Conclusions

Our research identified notable change in disease dynamics associated with implementation of mask mandates. Many factors will affect the future impacts of COVID-19, with vaccination expected to be the most critical. However, it is likely that COVID-19 will remain an important public health threat for the foreseeable future. Just as importantly, infectious disease experts caution that future pandemics remain a very real risk, with respiratory diseases being the most likely. The notable economic and social impacts of early "shutdowns" make clear that more targeted interventions are needed for future control efforts. Policy makers should consider the possibility that less intrusive measures, including mandating use of face masks, may be effective at minimizing disease spread while avoiding disruptive effects of less focus strategies.

## Supporting information

**S1 Fig. Graphic representation of parallel trends for municipalities that would adopt mandates and those without mandates.**
(JPG)

**S1 Data.**
(XLSX)

## Author Contributions

**Conceptualization:** Jared D. Taylor.

**Data curation:** Jared D. Taylor, Melinda H. McCann, Scott J. Richter, Dakota Matson, Jordan Robert.

**Formal analysis:** Melinda H. McCann, Scott J. Richter.

**Investigation:** Jared D. Taylor, Jordan Robert.

**Methodology:** Jared D. Taylor, Dakota Matson.

**Project administration:** Jared D. Taylor.

**Supervision:** Jared D. Taylor.

**Validation:** Dakota Matson, Jordan Robert.

**Writing – original draft:** Jared D. Taylor, Melinda H. McCann, Scott J. Richter.

**Writing – review & editing:** Jared D. Taylor, Melinda H. McCann, Scott J. Richter, Dakota Matson, Jordan Robert.

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
