## [Decision Letter · Decision Letter 0]

20 Jan 2022

PONE-D-21-39472Impact of local mask mandates upon COVID-19 case rates in OklahomaPLOS ONE

Dear Dr. Taylor,

Thank you for submitting your manuscript to PLOS ONE. After careful consideration, we feel that it has merit but does not fully meet PLOS ONE’s publication criteria as it currently stands. Therefore, we invite you to submit a revised version of the manuscript that addresses the points raised during the review process.

We look forward to receiving your revised manuscript.

Kind regards,

Sinan Kardeş, M.D.

Academic Editor

PLOS ONE

Journal Requirements:

"Unfunded study."

Reviewers' comments:

Reviewer's Responses to Questions

**Comments to the Author**

1. Is the manuscript technically sound, and do the data support the conclusions?

Reviewer #1: Yes

Reviewer #2: Partly

Reviewer #3: Yes

2. Has the statistical analysis been performed appropriately and rigorously? 

Reviewer #1: No

Reviewer #2: Yes

Reviewer #3: Yes

3. Have the authors made all data underlying the findings in their manuscript fully available?

Reviewer #1: Yes

Reviewer #2: No

Reviewer #3: No

4. Is the manuscript presented in an intelligible fashion and written in standard English?

Reviewer #1: Yes

Reviewer #2: Yes

Reviewer #3: Yes

5. Review Comments to the Author

Reviewer #1: 1. In lines 55 and 56 you said "Numerous studies have demonstrated the efficacy of face coverings in reducing transmission, but these have generally been in a healthcare setting", while in reference 6 efficacy of face coverings and other restrictions are compared to healthcare and non-healthcare setting. Please correct this.

2. From lines 106 to 129 you have provided a description of Table 1 that has no specific reference. Please provide a reference for this content. If the information was collected from a specific center (such as your center) you should mention this.

3. Please tell us, have you done an analysis to check for the presence of outliers in the statistical analysis section? In Figure 1, there appears to be an outlier value that affects your analysis output.

4. In lines 208, 238, 249, and 308, you said that seasons can be a risk factor for increasing the number of cases of Covid-19. It is not true. Please check "Pan, J., et al., Warmer weather unlikely to reduce the COVID-19 transmission: An ecological study in 202 locations in 8 countries. Sci Total Environ, 2021. 753: p. 142272."

5. Please indicate whether during your study, vaccination was in progress or not. If yes, you should not ignore the effect of vaccination.

Reviewer #2: The authors seek to contribute to the fight against the COVID-19 pandemic. Use of face masks or coverings is one of the major recommendations for reducing transmission, hence the relevance of this manuscript. Comments to be addressed to improve on the quality of the manuscript are as follows:

1. The methodology is not clear. A flow chart on the sample analyzed will bring some clarity. Also case counts for the period of analysis is needed.

2. Authors should provide literature/references on the enforcement of the mask mandate in Oklahoma. This will strengthen argument on the impact of the mandate.

3. Authors should provide the number of municipalities in Oklahoma and the exact number that implemented the mask mandate (39 or 40: refer to lines 110 and 160)

4. Why the selection of d-45 to d90 and not d-45 to d45 (providing equal intervals pre-and post-)?

5. Authors should provide the actual dates for period of analysis. Which date was d0?

6. Authors should check table 1:

- There are two columns for "Mandate start".

- Should also check dates for Midwest city.

- Implementation of state-wide restrictions (March-June 2020) is likely to dilute the impact of mask mandates. Authors should consider excluding the first five municipalities.

7. Authors should provide reference for the CDC classifications (lines 134-139)

8. There is no data or/and references in the manuscript to support lines 200-210)

9. Difficult to appreciate higher rate of increase in municipalities to adopt the mask mandate. What was peculiar about these municipalities? (lines 215 - 217). Authors need to describe the study sites in the methodology

10. One would have expected cases to increase faster in municipalities not implementing the mask mandate. This needs explanation (lines 218 -2020)

11. Authors should provide references for categorical statements (lines 280 - 282)

Reviewer #3: Please see the attached document for my review.

6. PLOS authors have the option to publish the peer review history of their article (what does this mean?). If published, this will include your full peer review and any attached files.

Reviewer #1: **Yes: **Mehrdad Bagherpour Kalo

Reviewer #2: No

Reviewer #3: No

---

## [Author Response · Author response to Decision Letter 0]

11 Mar 2022

As always, we are appreciative of the time and effort of the reviewers. We believe that the revised manuscript is notably improved by consideration of the suggestions and issues raised. We have endeavored to address all inquiries below, and have accepted/addressed nearly all suggestions or requests. The one suggestion that we are reluctant to incorporate is the graphic reporting of direct rates for day counts (as opposed to the difference in rate counts). We have concerns in doing so, which we ask the reviewers and editor to consider, but we would further evaluate such a figure if reviewers and editor deem it appropriate and beneficial. 

Thank you for your continued investment of time. We look forward to continued improvement and publication of this timely work.

Review Comments to the Author

Reviewer #1: 1. In lines 55 and 56 you said "Numerous studies have demonstrated the efficacy of face coverings in reducing transmission, but these have generally been in a healthcare setting", while in reference 6 efficacy of face coverings and other restrictions are compared to healthcare and non-healthcare setting. Please correct this. 

Done

2. From lines 106 to 129 you have provided a description of Table 1 that has no specific reference. Please provide a reference for this content. If the information was collected from a specific center (such as your center) you should mention this.

The process of identifying these municipalities was described in the methods. Table 1 has been moved from the introduction section into the results section to clarify the listing was created as part of the research process, and no reference for outside sources can be provided.

3. Please tell us, have you done an analysis to check for the presence of outliers in the statistical analysis section? In Figure 1, there appears to be an outlier value that affects your analysis output.

Yes, impact of outliers was assessed. It was found that none exerted disproportionate influence, and therefore all data points were retained in final results. Statements were added in the methods and results section to reflect this.

4. In lines 208, 238, 249, and 308, you said that seasons can be a risk factor for increasing the number of cases of Covid-19. It is not true. Please check "Pan, J., et al., Warmer weather unlikely to reduce the COVID-19 transmission: An ecological study in 202 locations in 8 countries. Sci Total Environ, 2021. 753: p. 142272."

Line 87 states “these waves have not necessarily followed seasonal [patterns]…,” reflecting uncertainty of the role of season. Line 213 (formerly line 208) does not assert season as a risk factor; it merely notes that cases increased in late fall and early winter. Figure 3 has been added to show the timeframe of case dynamics within the state.Similarly, line 242 (formerly 238) states “coincided roughly,” conveying both the factual occurrence but conceding uncertainty of true seasonality. Line 253 (formerly 249) was modified to reflect seasonality (or general changes over time) as a potential confounder. Line 320 (formerly 308) was expanded to acknowledge the conflicting information related to seasonality (the provided citation was included, as well as two additional references).

5. Please indicate whether during your study, vaccination was in progress or not. If yes, you should not ignore the effect of vaccination.

A paragraph was added in the discussion (line 291) addressing vaccine availability. 

Reviewer #2: The authors seek to contribute to the fight against the COVID-19 pandemic. Use of face masks or coverings is one of the major recommendations for reducing transmission, hence the relevance of this manuscript. Comments to be addressed to improve on the quality of the manuscript are as follows:

1. The methodology is not clear. A flow chart on the sample analyzed will bring some clarity. Also case counts for the period of analysis is needed.

An attempt at a flow chart of the methodology has been made and provided as figure 1. Case rate differences for each day count (which is what the actual analysis was based upon) will be provided in the supplemental materials. Raw case counts for each municipality cannot be provided due to data reporting restrictions.

2. Authors should provide literature/references on the enforcement of the mask mandate in Oklahoma. This will strengthen argument on the impact of the mandate.

Enforcement of mandates varied immensely geographically and over time. An explanation to this effect has been added (lines 271-275).

3. Authors should provide the number of municipalities in Oklahoma and the exact number that implemented the mask mandate (39 or 40: refer to lines 110 and 160)

The correct number is 39. Line 160 has been corrected. 

4. Why the selection of d-45 to d90 and not d-45 to d45 (providing equal intervals pre-and post-)?

Any interval was going to be artificial. We felt that -45 to 90 days gave a fairly comprehensive assessment for several reasons. 

a. 45 days would seem adequate to create a trend for pre-mandate implementation, and it would be unlikely that longer periods would bring more clarity. When policy makers were considering mandates, it is unlikely they would examine data from 60 days prior.

b. 45 days also minimizes the loss of data. Stretching farther back than 45 days would necessitate excluding municipalities that implemented mandates early in the pandemic.

c. 90 days would seem adequate to establish any trend post-implementation. If a mandate works, the benefit should be amplifying (i.e., the resulting reduced transmission rate at day 21 should further feed a reduction in transmission on days 28, 35, etc.). Because of this, we felt the post-mandate interval should be longer than the pre-mandate. 

d. That said, extending beyond 90 days would have again resulted in a loss of data, as many municipalities that implemented later mandates did not have data for even 90 days post-mandate. 

5. Authors should provide the actual dates for period of analysis. Which date was d0?

The specific date for D0 varied by municipality, depending upon when the mandate was implemented for that location. See the flow chart for further clarification. 

6. Authors should check table 1:

- There are two columns for "Mandate start".

- Should also check dates for Midwest city.

Thank you! Those errors were addressed

- Implementation of state-wide restrictions (March-June 2020) is likely to dilute the impact of mask mandates. Authors should consider excluding the first five municipalities.

It is possible that state-wide restrictions would alter case rates. However, any effect should be largely non-differential between communities with mandates and those without. Because the study analysis only examines differences in case rates, we believe that the effect of restrictions on our study should be minimal. Our approach chose to remove from the calculations any locations from all calculations after a mandate expired. Two of the early adopters expired rapidly, meaning they contributed only to the shortest of observed intervals and were not included in considerations beyond that point. The other three had much longer lived mandates, and they therefore contributed to the full length of examination. Removal of them from all analysis would reduce our sample size. Adding them in as D0 at a time after state-wide restrictions were lifted would misrepresent the effect of mandate over time. While your comment is appreciated, after consideration, we’ve decided it is preferable to not remove those early municipalities.

7. Authors should provide reference for the CDC classifications (lines 134-139)

Added.

8. There is no data or/and references in the manuscript to support lines 200-210)

Figure 2 has been added to show the dynamics of case rates throughout the state.

9. Difficult to appreciate higher rate of increase in municipalities to adopt the mask mandate. What was peculiar about these municipalities? (lines 215 - 217). Authors need to describe the study sites in the methodology

Our best efforts at explanation are offered in lines 233 through 240- the fact that more urban areas were more likely to adopt mask mandates. Given that there are 39 municipalities included in the study, and they are notably diverse in geography, demographics, and other features, it would be impossible to describe the sites in any satisfactory detail. 

10. One would have expected cases to increase faster in municipalities not implementing the mask mandate. This needs explanation (lines 218 -2020)

Our best explanation for this is again offered in lines 233-240. The municipalities that implemented mask mandates were more urban, and thus subject to generally higher rates of transmission. Mask mandates reduced much of the higher risk, but cannot necessarily be expected to completely reverse those dynamics. 

The study design was essentially a difference in differences method, whereby a difference was calculated (rate in mandated municipality minus rate in non-mandated population) and then changes in that difference was assessed over time (and specifically, before vs. after implementation). Thus, the comparison does not necessitate a faster increase in rates in non-mandate municipalities in order to detect an effect. Indeed, increases aren’t even necessary (as cases dropped at different times throughout the study). The real assessment is the difference between rate dynamics in mandate vs. non-mandate communities. 

11. Authors should provide references for categorical statements (lines 280 - 282)

Done

Overall

This article makes a valid contribution to the scientific record by evaluating the efficacy of local mask mandates in reducing COVID-19 case rates. The present work highlights an important limitation of previous mask mandate efficacy research and employs an analytic strategy that minimizes this limitation. Previous research has focused on comparing individual municipalities with themselves at two or more different points in time (i.e., prior to and following mask mandate implementation). These strategies are limited by their inability to largely account for temporal changes in other factors besides mask mandates that may influence COVID-19 transmission. To address this longitudinal limitation, the present manuscript instead compares mandated and non-mandated municipalities in Oklahoma at the same points in time. Specifically, a modified difference-in-difference analytic approach was used to compare the difference in case rates between 40 Oklahoma municipalities that eventually implemented a mask mandate and the rest of the non-mandated municipalities at the same points in time. The finding that locally-mandated mask mandates can mitigate differences between these two groups even when COVID-19 case rates are initially higher in the “eventual mandate” population than the “never mandate” population prior to implementation is very encouraging. 

While this work makes a substantive contribution to knowledge, some minor revisions are first required to improve readability, resolve conflicting or confusing statements, add a little bit of information to the Methods and Materials, fix in-text and Table errors, and explain/elaborate on certain statements.

Data Availability

I know the authors can’t share raw data owing to the protection of confidential medical information. However, the authors could provide the case rate for each municipality and non-mandate population at each d(-45) to d(90) time point. 

The case rate differences of rates in mask-mandate municipalities minus rates in non-mandated population of the state is provided in supplementary materials

Abstract

Lines 28-31:

In the abstract, the slope of difference estimates given in brackets may be difficult for the reader to understand since they have not yet read the paper. Word count permitting, I’d like to suggest the authors explain the first slope of difference in words. For example:

“Prior to adopting mask mandates, those municipalities that eventually adopted mandates had higher transmission rates than the rest of the state, with the mean case rate difference per 100,000 people increasing by 0.32 per unit increase in time (slope of difference= 0.32; 95% CI 0.13 to 29 0.51).”

If the authors cannot do this in the abstract, I’d recommend doing something similar in the results the first time an estimate for the slope of mean case rate difference is presented.

Done

Introduction

The authors provide an excellent summary of the COVID-19 timeline and mitigation efforts, a thorough background of existing research, a clear identification of research gaps, and a sound justification for the present work. However, the authors should consider reorganizing the summary of existing research to enhance reader clarity.

Attempts were made to improve the clarity and transitions between topics. If reorganizing is still justified, we would welcome specific suggestions (we may be too close to the subject to appreciate the issue!).

Line 48-49:

Could the authors briefly list some of the other “number of factors” besides “limited adoption” in this sentence (e.g., politicization of masks, economic burden, etc.)?

Done

Lines 51-104 (Paragraph 2):

This is a very long paragraph-- 2.5 pages! To help guide the reader, this text should be divided into multiple paragraphs that follow a clear, logical framework.

Done

Suggested breaks:

- End of line 71 seems like a good cut-off for a new paragraph. After this point, you transition into specific US studies and then focus on the “Joo, et al.” and “Lyu and Wehby” publications.

- Line 92 after “short periods of time” seems like a natural break because you stop talking about temporal factors and switch to other factors.

Line 73:

In this sentence it is unclear why “longitudinally” is not ideal. Perhaps the authors could consider adding a few words to help out the reader. For example:

“…the assessment was don’t longitudinally, which cannot account for xyz, and included…”

Done 

Lines 74-76:

You reference two papers for the first time and then refer to them by “Joo, et al.” and “Lyu and Wehby” only later in the paragraph. It would be clearer if you explicitly wrote “Joo, et al.” and “Lyu and Wehby” in this first sentence that introduces their work.

Done 

Lines 84-85 and 89-90:

The sentence “And while Joo…” is confusing with where it is located. The authors just finished discussing how including US states as a variable in a regression model requires the state effect to remain constant over time, and then the present sentence implies that having a seven-week study observation is a bad thing. This is confusing because the reader may presume (like I did) that having a shorter observation period = less opportunity for the effect to vary over time. Consider appending this sentence to the end of the “Regardless of causes(s)…”. For example:

 “…in any seven-week period, such as the seven-week period observed by Lyu and Wehby.”

For further resolution, the authors could consider explicitly contrasting the trade-offs of using longer observational periods in order to include the complexity of periodicity versus using a shorter observational period where the effect is more likely to be constant but periodicity is not captured.

Lines 101-104:

These sentences seem out of place because they go back to discussing variability over time. Consider moving to the end of line 92 after “…short periods of time”.

Extensive changes were made to these sections to improve clarity and flow of content.

Lines 121-123:

The authors saying that they used a modified difference-in-difference approach because of mandates going into effect at different times will likely not be self-explanatory to non-epidemiologists/statisticians. Perhaps a quick explanation as to why difference-in-differences is beneficial could be added at the end of the sentence to help out the reader. For example, a layman’s explanation of how the difference-in-differences approach does not assume exchangeability between the mandate and non-mandate groups. This could also be used to briefly justify not adjusting for the other factors discussed on the previous page (population density, socioeconomic status, political factors, etc.). 

Materials and Methods

Lines 132-134:

Since the information captured by state health departments likely varies by state, it may be beneficial to specify which events qualify for capture/what data sources the OK registry uses (i.e., specify the source population). Does the registry count COVID-19 cases in pharmacies, universities, assisted-care living facilities, etc., or just hospitals?

A couple sentences were added to clarify this. The authors are uncertain whether this information is more appropriately placed in the materials and methods, or in the discussion. We welcome comment on this matter.

Lines 150-153:

I think “…were calculated for each municipality to establish a mandate” is a typo. It reads as if the presence of a mandate was determined by municipality case density rates, which doesn’t make sense. I’m not sure what this means.

We apologize for the confusion. The sentence wasn’t intended to imply “… for each municipality IN ORDER TO establish a mandate…” Rather, it is meant as “…rates were calculated for municipalities that adopted mandates.” The sentence was modified to attempt to clarify.

Additionally:

Can the authors comment on the number/size of the aggregated non-mandate population? Just so we know there is an adequate non-mandate population size for comparisons with the mandated municipalities (which already have the sizes provided).

This was added on line 220

Results:

Line 186:

To support this sentence (i.e., the parallel assumption), perhaps the authors could supplement with a graph.

This has been added as a supplemental figure.

Line 189:

I was confused by this sentence because the preceding paragraph in the Methods and Materials said that there was significant autocorrelation. Specifying that this is for the one-lag moving average term model would improve clarity.

The p-value was moved to follow the statement that there was no remaining autocorrelation after addition of the one-lag moving average term, and the sentence was stricken from the results.

Lines 190-192:

- Can the authors comment on the parallel assumption not being violated despite the case rate difference rapidly increasing in the pre-mandate period (i.e., the case rate lines for mandated and non-mandated populations not being parallel since the difference between the two lines increases by 0.32 per unit time)? I had a hard time resolving these two pieces of information, which seem to contradict each other. Is it because the parallel assumption was done prior to date alignment, but the increasing case rate difference in on the aligned time scale?

- Can the authors explain the slope of 0.32 in words since this is the first time they present the slope of mean case rate difference (see comments for Lines 28-31)?

Additional explanation was added at the end of the results section.

Discussion:

Lines 234-236:

- This sentence is out of place. This concluding sentence (“The fact that…”) is sandwiched by two other sentences (rural versus urban areas and additional possible explanations) that aren’t directly related to it or support it. Instead, this sentence seems to interrupt the discussion about possible reasons for the difference. The authors should consider relocating it.

The sentence was moved to earlier in the same paragraph, to speak more generally about rural vs. urban, and to not interrupt the specific comments about the state of Oklahoma.

Lines 236-243:

- These sentences are confusing. I think “… subsequent increase in case rate difference between communities with mask mandates and those without” indicates there is an increase in the case rate difference after the mandate. This isn’t communicated in the results (which say there is a decrease), so at first I thought I was misinterpreting this sentence. However, the following sentences hypothesize why the mask mandates don’t work as well over time following the mandate (temperature, gradual non-compliance), which makes me think that the “… subsequent increase” sentence is indeed about post-mandate periods of time. Then, the last sentence talks about a spillover effect with mandates benefitting nearby non-mandate regions by reducing transmission to them. I don’t understand how this could explain an increase in the case rate difference—wouldn’t this transmission reduction decrease the case rate difference? 

- I’m not sure if something is missing from the results or this part of the discussion needs work or shuffling to improve clarity. Presently, it is difficult read and understand.

The authors agree that the statement was poorly constructed and difficult to understand. Lines 279-282 have been rewritten in attempt to more effectively communicate the findings. Critically, the phrase “subsequent increase in” was replaced by “persistently higher.” The statement “increase in” was factually incorrect. However, it is important to note that, despite the fact that the slope reversed course, the mean difference remained above zero, even 90 days after mandate implementation. This means that case rates in mandated communities still exceeded those in non-mandated communities. However, the difference was shrinking, and may have eventually reached equality.

Line 246:

Change “compare” to “compared”.

Done

Line 247:

Is aligning time to d0 solely responsible for avoiding confounding? Wouldn’t calculating and comparing non-mandate rates with mandate rates on the same calendar date be extremely important for avoiding confounding by temporal factors? I think it is important to mention this, otherwise it reads as if d0 is solely responsible for preventing confounding. In fact, I’m not entirely clear on how d0 avoids temporal confounding when the rate differences for a particular city and the non-mandated population are always calculated on the same day—would it not be this technique of using the same day for the mandated city and non-mandated population comparison that removes confounding and not aligning to d0?

The sentence has been changed to state that it is both the use of differences in rates, and the use of D0, that avoids the risk of confounding. Utilizing the difference of rates largely eliminates risks of confounding by season, etc., because this essentially creates a comparison by date. However, given that implementation dates varied immensely, setting to D0 permits consolidation of municipalities that adopted mandates at radically different times into a single assessment. Because it is reasonable to think that effects of a mandate compound, there should be a linear relationship of declining rates with longer period of mandate implementation. This could not be assessed without setting to d0. 

Lines 270-277

It is very important for us to remember that observed COVID rates rely on the availability of widespread testing. The authors explain this issue very well and then succinctly describe how/why this bias is limited in their study.

Thank you!

Lines 280-283:

It is not explicitly clear why high positivity rate implicate a large number of infections not being detected. Could the authors briefly elaborate?

Additional discussion was added

Lines 287-300:

This is an excellently written and technically sound justification for why adjustment for other rate-influencing covariates was not performed.

Thank you!

Line 313:

Change “by able” to “be able”. 

Done

Tables and Figures:

Table 1:

I think there is a typo in the rightmost column of Table 1. I think this is supposed to be “Mandate Expiration” and not “Mandate Start”.

Fixed

Addition of Figure:

To improve the reader’s understanding of the results, the authors could consider supplementing an additional figure with:

- Y-axis: mean case rate per 100,000

- X-axis: alignment (d(-45) to d(90))

- One line for case rates in the non-mandated population

- A second line for case rates in the mandated population

- Perhaps mean case rate data points for each population (mandated and non-mandated) at each time point

This will allow the reader to observe the case rates over time for the mandated vs non-mandated groups. Changes in case rates in two populations over time are less abstract and more intuitively understood than changes in case rate differences over time. The possible inclusion of the suggested case rate graph would help non-statistician/epidemiologist readers understand the results.

From a statistical perspective, we are reluctant to provide such a graphic because the population for the non-mandate denominator would be undefined (i.e., each person is not represented once and only once for each calculation). We are also hesitant to provide it from a non-statistical perspective because it may create confusion as to the actual method of analysis. We would consider the possibility further if the reviewer considers it an imperative point. 

Figure 1:

- Both the regression lines and the data points in Figure 1 are plotted with circles of the same shape. When I first reviewed the figure, I thought the regression lines were also data points. Since the graph is in gray-scale, different shapes/symbols should be used for the regression lines and data points.

- It is not immediately clear what the data points (not lines) are in Figure 1. I think, but am not sure, that each point is the average of the 40 separate differences in case rates between mandated and non-mandated municipalities for each time point. This should be explicitly specified in the Figure title. Better yet, add a legend explaining what the individual data points and regression lines are.

- The methods describe calculating the difference in rates from d(-45) to d(90), but Figure 1 only shows up until d(75) instead of d(90). Is there an error in the axis? If not, perhaps the authors could briefly mention why only up until d(75) is shown. 

All of the described issues have been addressed.

---

## [Decision Letter · Decision Letter 1]

18 Apr 2022

PONE-D-21-39472R1Impact of local mask mandates upon COVID-19 case rates in OklahomaPLOS ONE

Dear Dr. Taylor,

Thank you for submitting your manuscript to PLOS ONE. After careful consideration, we feel that it has merit but does not fully meet PLOS ONE’s publication criteria as it currently stands. Therefore, we invite you to submit a revised version of the manuscript that addresses the points raised during the review process.

We look forward to receiving your revised manuscript.

Kind regards,

Sinan Kardeş, M.D.

Academic Editor

PLOS ONE

Journal Requirements:

Reviewers' comments:

Reviewer's Responses to Questions

**Comments to the Author**

1. If the authors have adequately addressed your comments raised in a previous round of review and you feel that this manuscript is now acceptable for publication, you may indicate that here to bypass the “Comments to the Author” section, enter your conflict of interest statement in the “Confidential to Editor” section, and submit your "Accept" recommendation.

Reviewer #1: All comments have been addressed

Reviewer #2: (No Response)

2. Is the manuscript technically sound, and do the data support the conclusions?

Reviewer #1: Yes

Reviewer #2: Yes

3. Has the statistical analysis been performed appropriately and rigorously? 

Reviewer #1: Yes

Reviewer #2: Yes

4. Have the authors made all data underlying the findings in their manuscript fully available?

Reviewer #1: Yes

Reviewer #2: Yes

5. Is the manuscript presented in an intelligible fashion and written in standard English?

Reviewer #1: Yes

Reviewer #2: Yes

6. Review Comments to the Author

Reviewer #1: (No Response)

Reviewer #2: Authors have largely addressed previous comments. A few issues with figure 1 - a bit crowded:

Step 1: Authors should specify the dates. Is it 15th February 2020 to 6th February 2021 (from figure 2) or 17th March 2020 to 1st March 2021 (from materials and methods section)?

Step 2: Is this number of cases for days or dates? If days, then consider revising to ".....each day (d-45 to d90)........"

Authors can have an arrow to indicate what d0 is (i.e. d0=day of mask mandate implementation). Authors will not need 3a if the suggestion is adopted.

Step 3: Consider revising to "........calculate daily rates (cases.....) for each community". As indicated earlier, 3a becomes redundant.

Step 4: Consider revising to "............case count to get sum of cases from municipalities without mask mandate"

Step 5: Not fully visible. Revising step 4 may make step 5 redundant, and could be deleted.

Step 6. Consider revising to "...single daily "non-mask mandate" rate (.....) calculated for the state"

Step 7: Consider revising to "Daily differences in rates were calculated for each mandate municipality: (.......)"

Steps 8 - 10: Appear redundant if the above suggestions are adopted, and could be deleted.

Step 11: Consider revising to "Mean rates calculated for each day"

7. PLOS authors have the option to publish the peer review history of their article (what does this mean?). If published, this will include your full peer review and any attached files.

Reviewer #1: No

Reviewer #2: No

---

## [Author Response · Author response to Decision Letter 1]

22 Apr 2022

Response to editor: 

Comment was made in the review regarding citation of retracted articles. I am unclear if this is a standard request or a specific citation raised concern. All citations were verified as having no retractions. One article had an erratum issued in a subsequent publication. Information on this has been added to the references section. This addition is the only change made to the manuscript itself. The only other changes requested were in relation to figures 1 and 3. 

Response to reviewers:

The authors are again appreciative of the time and expertise of the reviewers. No comments were provided from reviewer #1 for the resubmission, and thus no actions were taken. Reviewer #2 provided suggestions for figure #1, many of which were adopted. It is the authors’ opinion that other recommendations (specifically, changing the reference of dates to days) would misrepresent the data analysis process. For Step 2, the reviewer asked: “Is this number of cases for days or dates? If days, then consider revising to ‘…..each day (d-45 to d90)…..” The number at that step is for specific dates. The conversion to days does not occur until after the difference has been calculated for each municipality (i.e., rate for municipality - rate in remainder of the state), which is shown as step 8. The data analysis had to be done in this order was because there is no single “d0” rate for the non-mandated portion of the state. Conceptually, the figure could be altered to state that the differences were calculated for each day count without necessarily changing the interpretation. This would indeed simplify the figure. However, since this is not consistent with the actual method of data analysis, the authors are reluctant to adopt such changes. 

Figure 3 was updated with truncation of data to be consistent with intervals of the study period.

---

## [Decision Letter · Decision Letter 2]

19 May 2022

Impact of local mask mandates upon COVID-19 case rates in Oklahoma

PONE-D-21-39472R2

Dear Dr. Taylor,

We’re pleased to inform you that your manuscript has been judged scientifically suitable for publication and will be formally accepted for publication once it meets all outstanding technical requirements.

Kind regards,

Sinan Kardeş, M.D.

Academic Editor

PLOS ONE

Additional Editor Comments (optional):

Reviewers' comments:

Reviewer's Responses to Questions

**Comments to the Author**

1. If the authors have adequately addressed your comments raised in a previous round of review and you feel that this manuscript is now acceptable for publication, you may indicate that here to bypass the “Comments to the Author” section, enter your conflict of interest statement in the “Confidential to Editor” section, and submit your "Accept" recommendation.

Reviewer #1: All comments have been addressed

Reviewer #2: All comments have been addressed

2. Is the manuscript technically sound, and do the data support the conclusions?

Reviewer #1: Yes

Reviewer #2: Yes

3. Has the statistical analysis been performed appropriately and rigorously? 

Reviewer #1: Yes

Reviewer #2: Yes

4. Have the authors made all data underlying the findings in their manuscript fully available?

Reviewer #1: Yes

Reviewer #2: Yes

5. Is the manuscript presented in an intelligible fashion and written in standard English?

Reviewer #1: Yes

Reviewer #2: Yes

6. Review Comments to the Author

Reviewer #1: (No Response)

Reviewer #2: (No Response)

7. PLOS authors have the option to publish the peer review history of their article (what does this mean?). If published, this will include your full peer review and any attached files.

Reviewer #1: No

Reviewer #2: No

---

## [Editor Report · Acceptance letter]

8 Jun 2022

PONE-D-21-39472R2 

Impact of local mask mandates upon COVID-19 case rates in Oklahoma 

Dear Dr. Taylor:

I'm pleased to inform you that your manuscript has been deemed suitable for publication in PLOS ONE. Congratulations! Your manuscript is now with our production department. 

Kind regards, 

on behalf of

Dr. Sinan Kardeş 

Academic Editor

PLOS ONE